# The Influence of WEDM Parameters Setup on the Occurrence of Defects When Machining Hardox 400 Steel

**DOI:** 10.3390/ma12223758

**Published:** 2019-11-15

**Authors:** Katerina Mouralova, Tomas Prokes, Libor Benes, Josef Bednar

**Affiliations:** 1Faculty of Mechanical Engineering, Brno University of Technology, 616 69 Brno and Czech Republic; mouralova@fme.vutbr.cz (K.M.); bednar@fme.vutbr.cz (J.B.); 2Faculty of Production Technologies and Management, Jan Evangelista Purkyně University, 400 96 Ústí nad Labem, Czech Republic; libor.benes@ujep.cz

**Keywords:** WEDM, electrical discharge machining, hardox, cavities, cracks

## Abstract

The unconventional technology wire electrical discharge machining is a highly used technology for producing precise and indented shaped parts of all materials that are at least electrically conductive. Its wide use makes this technology applicable in almost all branches of industry, even in the automotive industry, where the abrasion resistant material under investigation Hardox 400 steel is widely used for the manufacturing of truck bodies. The aim of this study was a comprehensive analysis of the machinability of this material using WEDM employing a 33−round experiment. Based on the change in machine parameters (pulse off time, gap voltage, discharge current, pulse on time, and wire feed), the cutting speed, the topography of machined surfaces, and the chemical composition of the workpiece surface, the morphology and condition of the subsurface layer including lamella production and a subsequent determination of the distribution of individual elements in the given area were analyzed. It has been found that during the machining of this steel, many defects occur in the subsurface layer of the material in the form of cracks with a depth of up to 22 µm and burned cavities. However, by appropriately adjusting the machine parameters, it was possible to completely remove these cracks.

## 1. Introduction

Wire electrical discharge machining (WEDM) is an unconventional machining technology that utilizes electrical impulses generated by a generator to remove a material very efficiently. These periodically repetitive impulses cause the erosion of the workpiece material (as well as the used wire electrodes in part), which is subsequently flushed from the cut point with a dielectric fluid stream. In addition to erosion, the workpiece material is also removed due to high temperatures (10,000–20,000 °C [1]), which cause its evaporation. The tool used here is a wire electrode that has a diameter of 0.02 to 0.3 mm and is made of brass, copper, molybdenum, or may be composite with optional coating [2,3].

Despite its energy intensity, WEDM is an irreplaceable technology in many industries, such as the automotive, aerospace, medical, and military industries. The abrasion−resistant Hardox 400 steel is a very important construction material because it has enabled the use of non−reinforced structures for truck bodies. Furthermore, it is used for the production of crushers, gears, containers, metal scrap processing machines, conveyors, cutting knives, etc. Hardox has a breaking strength of 1250 MPa, a yield strength of 1000 MPa, and a hardness in the range of 370 up to 430 Brinell hardness (HBW) [4,5].

Manjaiah et al. [6] studied the effect of process parameters on the material removal rate (MRR) and surface roughness (Ra) in WEDM of AISI D2 steel with the pulse on time and the servo voltage being the most significant parameters affecting MRR and Ra. The experiments were performed with different cutting conditions using the Taguchi based utility approach. The results showed that the machined surface hardness of steel D2 was increased due to the repetitive quenching effect and formation oxides on the machined surface. Lusi et al. [7] held experiments in order to determine the setting parameters of an appropriate WEDM process (such as the arc on time, open voltage, on time, servo voltage and off time) to maximize the material removal rate and minimize kerf ad surface roughness of the workpiece material, based on an orthogonal matrix L18 design. The results of the experiments showed that the setting parameters for the arc to on time, open voltage, and off time had the greatest contribution in reducing the variation of responses. Klocke et al. [8] performed an analysis of the white layer that could significantly affect mechanical behavior. They also investigated the structural constitution of the microstructure with the help of chemical analyses of the rim zone together with TEM−images; and the influence of the local defects and transformations on the workpiece functionality. The analysis performed helps to predict the mechanical behavior of workpieces machined with WEDM, especially for components with high ratios and filigree geometries. Abdullah et al. [9] focused on the study of the encouragement of three machining parameters on WEDM consisting of peak current, pulse off time, and wire tension to three machining responses, namely the cutting rate, material removal rate, and surface roughness on high carbon steel. The results showed that the relation between material removal rate and surface roughness indicate that the optimum machine parameter using brass wire as the electrode is C1. The spark analysis using brass wire showed that the deeper the spark occurred, the higher value of the surface roughness became. Reddy et al. [10] tried to determine the parametric influence on WEDM utilizing the Taguchi Technique. They focused on the estimation of surface roughness and volumetric material removal rate, with in bed speed, pulse on duration, pulse off duration and current as input parameters. The results of the experiments revealed that the overall performance is dependent on the pulse duration, current, pulse on, and bed speed in the order. Sinha et al. [11] implied in their experiments that a hybrid approach of the Taguchi method and principal component analysis for multi−objective optimization WEDM of tool steel AISI D3 could achieve better cut qualities within existing resources. They considered three quality characteristics of the material removal rate: the length of cut, surface roughness and the input parameters, namely pulse on time, pulse off time, peak current and servo voltage. The results of the performed optimization include the prediction of an optimum input parameter level and their relative significance on multiple quality characteristics. Nayak et al. [12] also employed a multi response optimization approach in their study in order to determine the optimal process parameters in the WEDM process during taper cutting operation. For the experiments performed six process parameters were used, such as part thickness, wire tension, taper angle, discharge current, pulse duration and wire speed, which were utilized at three levels to obtain the responses: angular error, surface roughness and cutting speed. 

WEDM is a phenomenon in which it is necessary to ensure excellent surface and subsurface quality, as well as the desired machining efficiency. For this reason, extensive research was carried out, including analysis of the influence of the cut orientation of the semi−product on the occurrence of cracks [13], the influence of the cut gap width on the accuracy of the machined parts depending on their heat treatment [14], analysis of the surface and subsurface layer quality for the aluminum alloy 7475−T7351 [15] and the pure aluminum material [16]. The purpose of this study was a comprehensive analysis of the topography of the machined surface of Hardox 400 after WEDM and a detailed analysis of the surface and subsurface layers after this type of machining. The machining of this abrasion−resistant steel with WEDM has not been studied in any study yet, despite the fact that its use in today’s industry continues to grow. The knowledge of the state of the surface and subsurface layers of the workpieces is essential, especially depending on the required service life of the parts produced.

## 2. Experimental Setup and Material

### 2.1. Experimental Material

The samples for the experiment were made of the wear−resistant Hardox 400 steel, which has the chemical composition according to the standard in weight percent: 1.5% Ni, 0.3% C, Mn, 1.4%, 0.7% Si, 1.6% Cr, 0.6% Mo, max 0.025% P, max 0.01% S, Fe −balance. This steel has a good cold flexibility, is abrasion−resistant, is well weldable, and its toughness is KV = 45 J. Hardox has a breaking strength of 1250 MPa, a hardness in the range of 370 up to 430 HBW, and a yield strength of 1000 MPa. It is used primarily for the manufacture of structures because it has good mechanical properties and is mainly abrasion−resistant. Furthermore, it is used for the production of crushers, containers, gears, conveyors, cutting knives, metal scrap processing machines, etc. A 15 mm thick prism semi−product was used for the experiment. The microstructure of this steel Hardox 400 is bainitic−martensitic and is shown in Figure 1 together with sample of produced specimens.

### 2.2. WEDM Machine Setup

A WEDM cutter EU64 from MAKINO was used for the machining. The machine was equipped with CNC control in 5 axes, and non−ionized water was used as a dielectric liquid. The wire electrode marked PENTA CUT E had a diameter of 0.25 mm and was made of 60% of Cu and 40% of Zn.

A Design of Experiment (DoE) is a test or more tests when the process input parameters systematically are changed to determine the corresponding changes in the output variable, the so−called response. For the response modeling the regression analysis and variance analysis (ANOVA) are used. With the help of DoE, the significance of predictors and their settings can be defined, which enables us to optimize the process. DoE was based on the monitoring the influence of five independent technological parameters of the cutting process, such as the pulse off time (*T_off_*), gap voltage (*U*), discharge current (*I*), wire feed (*v*) pulse on time (*T_on_*) and their limit values (Table 1). The limit values or each parameter setup were based on vast previous tests as well as the recommendations of the machine manufacturer.

For the experiment, “Half response surface design” was chosen, which contains 33 rounds separated into two blocks (Table 2). To reduce the possibility of systematic errors, each round was randomized and seven central points were added for the improvement of the statistical variability of response. This data collection plan is described in detail in, for example, Montgomery [17].

Unconventional WEDM technology also differs from common conventional technologies due to its machining speed or cutting control. WEDM does not allow a direct adjustment of the cutting speed *v_c_* when programming the machine but the speed is based on the setting of individual machine parameters. The cutting speed and the number of wire electrode breaking (the number was recorded on each separate sample, with the cut length of 3 mm each, as shown in Figure 1), make the resulting total machining time. The electrode breakage happens because of the inappropriate machine parameters setup. The rewiring of the wire electrode lasts 1 min, so a frequent breaking is not desirable and efficient for the machining process. The WEDM cutter used allowed a direct speed measurement during the machining process.

The individual cutting speed values of the produced samples were plotted in the graph shown in Figure 2. The highest speed was achieved for Sample 14, i.e., 4.1 mm·min^−1^, however, there was a single wire electrode breakage from all the machined samples. For Sample 31 that was cut at the second highest speed of 4 mm·min^−1^, no breakage occurred, so it can be said that the fastest machining will be achieved with the machine parameter setting *U =* 50 V, *T_on_* = 10 µs, *T_off_* = 30 µs, *v* = 14 m·min^−1^, and I = 35 A. In the following chapter, the optimization of machine setting parameters was made in order to maximize cutting speed in order to increase machining efficiency in the form of reduced machining time.

## 3. Statistical Evaluation of the Cutting Speed

Using the regression analysis, a mathematical model was created that allowed the calculation of the cutting speed *v_c_* using statistically significant parameters. The significance level chosen was 5%, with the model being developed using the “stepwise” method to select statistically significant parameters.

The determination coefficient R_2_ = 93.31%, thus, a model for the cutting speed expresses 93.31% of the variability of the measured cutting speed *v_c_*. The significant factors were pulse off time, discharge current, pulse on time and pulse off time wire feed interaction. Due to the model hierarchy, the wire feed factor has not been removed because has a significant interaction with the pulse off time.

The gap voltage factor is insignificant, including interactions and quadratic terms. The mathematical record of the model is then given in relation (1) and displayed in the contour plot in Figure 3a,b.
(1)$$v_{c}=3.836+0.1333T_{on}-0.0933T_{off}-0.1444v+0.05889I+0.00375T_{off}\cdot v$$

From the Main effect plot shown in Figure 3b the positive effect of discharge current, pulse on time and negative effect of pulse off time on cutting speed is shown. The wire feed factor is insignificant but has been included in the model due to a significant interaction.

## 4. Analysis of the Machined Surface and Discussion

### 4.1. Experimental Methods 

All experimentally produced samples were cleaned and analysed using a LYRA3 scanning electron microscope (SEM) from Tescan. This equipment was supplied with an energy−dispersive X−ray detector (EDX). In order to study the surface and subsurface microstructural changes, metallographic specimens showed that cross sections of each sample were produced. These metallographic specimens were prepared using conventional techniques: wet grinding and diamond paste polishing with a TEGRAMIN 30 automatic preparation system from Struers. The final mechanical−chemical polishing was done using the OP−Chem suspension from Struers. After the etching, the structure of the material was monitored and documented by electron and light microscope on the inverted light microscope (LM) Axio Observer 21m from ZEISS. Surface topography, planar, profile, and supporting profile parameters were further examined employing a non−contact 3D profilometer Taylor Hobson Talysurf CCI Lite. The measured data were afterwards subsequently processed in TalyMap Gold software, which enabled the formation of a 2D and 3D model of the analysed surface. 3D surface reliefs were further researched using the semi−contact technique Atomic force microscopy (AFM), and the measured data were analysed using the Gwyddion program. Using a focused ion beam (FIB) on a Helios microscope from FEI, the lamella was prepared to study the material composition using EDX in transmission electron microscope (TEM) Titan from FEI.

### 4.2. Surface Topography Analysis

Typically, the topography of the machined surface is carefully monitored because it must meet the values prescribed in the manufacturing documentation of the individual parts. The expected surface quality values are usually prescribed by the machine tool manufacturer only in the form of arithmetical mean deviation of profile (Ra) for the production technologies for individual materials and their thickness. However, there are many materials that are not included in these manufacturing technologies and Hardox is one of these examples. The analysis of the surface topography following the machine parameters setup is therefore necessary, especially when the part is only machined with WEDM without further finishing (e.g., grinding). For this reason, three parameters of the basic profile, three profile parameters, and their three surface equivalents were evaluated in this experiment, which provide quantitative evaluation of the area by all technically significant directions [18]. The basic profile parameters evaluated were Pa, Pz, and Pq. The parameters evaluated by the profile method were Ra, Rz, and Rq. The arithmetical Sa, Sz and Sq were evaluated by a planar method. All parameters were evaluated using the non−contact 3D profilometer Taylor Hobson Talysurf CCI Lite according to the corresponding standard for the surface parameters ISO 25178−2 [19] and profile ISO 4287 [20]. All parameters were evaluated on 1024 profiles of a single evaluation length Profile Length Ratio l = 0.8 mm obtained from S−F surfaces of measurements made with the 20× objective. Five random spots on each sample were selected for the measurement, followed by an average of these values.

The evaluated topography parameters of the individual samples were plotted in the graphs shown in Figure 4. The lowest values for almost all evaluated parameters (except for the parameter Sz) were obtained for Sample 30, which was machined with the machine parameters setup: U = 70 V, T_on_ = 6 µs, T_off_ = 30 µs, h = 14 m·min^−1^, and I = 35 A. This sample had a Ra value of 2.08 µm, which is significantly lower than that of Atlug research [21], but with different heat treatments for this steel. In contrast, the worst surface quality was evaluated for Sample 25, where all parameters except for parameter Sz reached the highest values. From this point of view, the parameter Sz, which can be defined as the sum of the highest value of the projection height and the highest depth value of the depression in the delimited area, appears to be a parameter reflecting the eventual occurrence of isolated protrusions resulting from the adherence of the discharged material particles on the machined surface.

To illustrate the relief of the machined surfaces, a color−filtered sample surface scan with the highest surface quality and the lowest was produced on the Taylor Hobson Talysurf CCI Lite. Both of these reliefs are shown in Figure 5 with a clear difference in their topography and height differences between individual protrusions and depressions, i.e. craters. Further, from the selected area on the machined surface of Sample 30 (with the highest surface quality) a relief was created using the semi−contact AFM technique, which is based on the detection of changes in interaction forces between the tip and the workpiece surface with a change of tip distance from the surface. The measurements were performed using the tip with a diameter of 0.65 µm in the Scanasyst mode. The evaluated area was 30 × 30 µm and is shown in Figure 5c. 

### 4.3. Surface Area Analysis

During the material removal process along with WEDM, a very specific morphology of the machined surfaces is generated by the action of the individual electrical impulses. This morphology is created by a large number of individual craters formed by the debris of small particles of material that were subsequently washed away by a dielectric fluid stream. However, the appearance of individual morphologies is different and is influenced not only by the set of mechanical and physical properties of the machined material [22], including the type of the heat treatment, but also by the direction of the cut of the semi−product [23] and, last but not least, by the setup of the machine parameters [15]. On the surface of the machined samples, due to the effects of very high temperatures, a recast layer is formed, which represents a layer of completely molten and re−cooled material.

The surface morphology and possible surface or subsurface defects are the key parameters determining the final quality of the machined surface. The study of the surface morphology of all machined samples was performed by electron microscopy. In all cases, a BSE backscatter detector was used for imaging, and the samples were always studied at a magnification of 1000×, 2500× and then 4000×. The appearance of the morphology of Sample 30 with the lowest roughness values is shown in Figure 6a, showing a large number of tiny black dots in the image, which represent tiny holes, which are more visible in the detailed image. The surface is relatively smooth, especially when compared to the surface of Sample 25 (the highest roughness values) shown in Figure 6b. This and almost all other produced samples are covered with a recast layer formed by a structure that is specific to special types of steels such as Hadfield [24].

Due to the very high temperatures, massive diffusion processes between the wire electrode and the workpiece material occur during the WEDM process, which has been presented in connection with the processing of various materials such as martensitic stainless steel [25], ASP 23 steel [26], aluminum alloys 7475−T7351 [15] or Inconel 625 [23], however, the machine parameter setup was found to have no diffusion and the workpiece surface was not contaminated with wire electrode elements (copper and zinc). With this presented research, it can be said that diffusion contamination can only be completely eliminated by optimizing the machine parameters setup. Furthermore, no diffusion processes in the machining of highly oriented pyrolytic graphite (HOPG) [27] were also studied. Figure 7 shows the individual places of the chemical composition analysis on the surfaces of Sample 25 (sample with the highest roughness values) and 30 (sample with the lowest roughness values), with the measurements showing that the individual formations on the surface 25 are created mostly of copper and zinc diffused from the wire electrode than the individual places examined on Sample 30. Most of copper and zinc were measured on Place 1 (Figure 7b) but this value was only 5.4 wt.% of copper and 3.4 wt.% of zinc. In the other three measured places, the occurrence of copper was always below 2.5 wt.% of copper and 1.8 wt.% of zinc. Another interesting place of measurement (Place 2) shown in Figure 7a was an adherent globule of debris, composed mainly of iron. In contrast, Place 1 is a visibly oxidized place because the oxygen content of 33.2 wt.% was detected here.

### 4.4. The Subsurface Analysis

The subsurface layer analysis was performed using electron microscopy on the pre−prepared metallographic specimens of all samples. This analysis is very important, since after WEDM, the defects in the form of cracks [13] or burnt cavities [24] are often found in the subsurface area, which can affect the correct functionality and durability of the part produced. The cross sections were observed using a Lyra3 scanning electron microscope. A BSE detector was used throughout the whole analysis, first with a magnification of 1000× and then with 2500× and 4000×.

The cross section analysis showed an occurrence of a large number of defects, both in the form of cracks and burnt cavities. The formation of burnt cavities is caused by high temperatures at the point of cut, with water dissociation in the water dielectric bath of the machine and the diffusion of atomic hydrogen below the surface of the machined material. These defects are shown in Figure 8, including their dimensions that range within microns. The longest crack had a length of 22 µm and is shown in Figure 8e. Longer cracks have only been studied with WEDM of pure molybdenum [28]. Obviously, these defects will further disrupt the integrity of the surface layer and may also cause the initialization of much larger cracks across the entire produced sample. The only sample on which only a few small burnt cavities were found is Sample 30, whose subsurface area is shown in Figure 9. This sample also shows that the recast layer is significantly thinner (max up to 5 µm) compared to all other samples, and there is no other on 80% of the entire sample surface. Other specimens have a recast layer of up to 20 µm on their entire surface.

### 4.5. The Analysis of TEM Lamella

The Helios electron microscope was used for the production of TEM lamella from Sample 25, with four basic steps of manufacturing. These are the deposition of a protective layer, the formation of a coarse lamella by means of an ion beam, the transfer to a copper holder, and the final thinning. The transmission microscope Titan Themis 60–300 was used for the observation of the produced lamella. The observation was performed at 300 kV of acceleration voltage and the beam current used during the EDX measurement was set to 1nA (was run in a scanning mode). At this setting, the volume size was 0.5 angstrom.

The chemical composition analysis of the prepared TEM showed a homogeneous distribution of elements in the base material, with the recast layer being the only area with different element concentrations. Figure 10a shows the produced lamella and the EDX analysis area, which was shown in the form of individual element distribution maps in Figure 10b. A smaller area than the entire area of the lamella was chosen to increase the accuracy of the individual elements. It is apparent from the EDX analysis that in the recast layer the base material was mixed with the electrode material, i.e. copper and zinc. In addition, the local concentration of the chemical elements of the base material in the recast layer has also changed. The concentration was increased in the elements: manganese, chromium, carbon, silicon, molybdenum, and nickel except for iron. For silicon, copper and manganese, the highest concentration was detected at the interface of the recast layer and the base material. A small amount of the observed copper on the surface of the lamella is caused by the secondary transfer of the material from the recast layer during the dedusting.

## 5. Conclusions

Based on the Design of the Experiment, 33 samples were produced and were used to minitor the influence of the machine parameters setup on the cutting speed and the resulting quality of the surface and subsurface layers. By analysing all the samples produced, the following conclusions were reached:Sample 31 was machined at the highest speed of 4 mm·min^−1^ without breaking of the wire electrode with machine setting parameters: *U* = 50 V, *T_on_* = 10 µs, *T_off_* = 30 µs, *v* = 14 m·min^−1^ and *I* = 35 A,using a regression analysis, a mathematical model was created to calculate the cutting speed, with a positive effect of the factor *T_on_*, I and the negative effect of *T_off_* on the cutting speed, based on the Main effect plot,the lowest roughness parameter values were obtained for Sample 30, i.e., only Ra 2.08 µm, (by machine parameters setup: *U* = 70 V, *T_on_* = 6 µs, *T_off_* = 30 µs, *v* = 14 m·min^−1^, and *I* = 35 A), and this pattern also had a visibly smoother 3D relief surface,the surface morphology of Sample 30 with the lowest roughness was covered by a noticeably smaller amount of recast layer than all other samples, and there was also significantly less diffusion of elements from the wire electrode,the subsurface analysis revealed a large number of cracks up to 22 µm in length and burned cavities in all samples produced, except for Sample 30, whose surface was covered with only a small amount of small cavities,the recast layer thickness for all samples examined was below 20 µm and covered the entire surface of each sample except for Sample 30, which was only covered by the recast layer by 20%, and the thickness did not exceed 5 µm,the analysis of TEM lamellae showed a homogeneous distribution of elements in the base material, with the recast layer as the only area with different concentration of elements.

Taking into account the above−mentioned conclusions, it can be unambiguously stated that the improvement of the surface and subsurface layers of Hardox 400 after WEDM can be achieved by setting the machine parameters to: *U* = 70 V, *T_on_* = 6 µs, *T_off_* = 30 µs, *v* = 14 m·min^−1^ and *I* = 35 A. Using this setting, it is possible to use a machine on a surface with a roughness of 2.08 µm and only a small amount of burned cavities. This setting will extend the life of the part and ensure that it functions properly.

## Figures and Tables

**Figure 1 materials-12-03758-f001:**
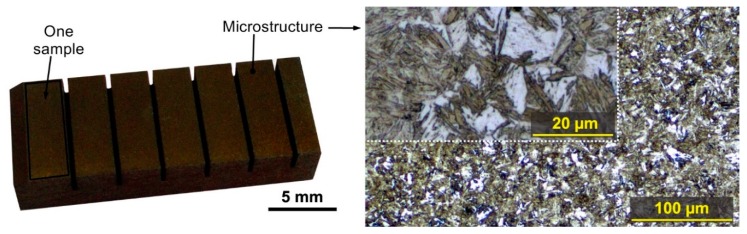
Experimental samples and their microstructure representation.

**Figure 2 materials-12-03758-f002:**
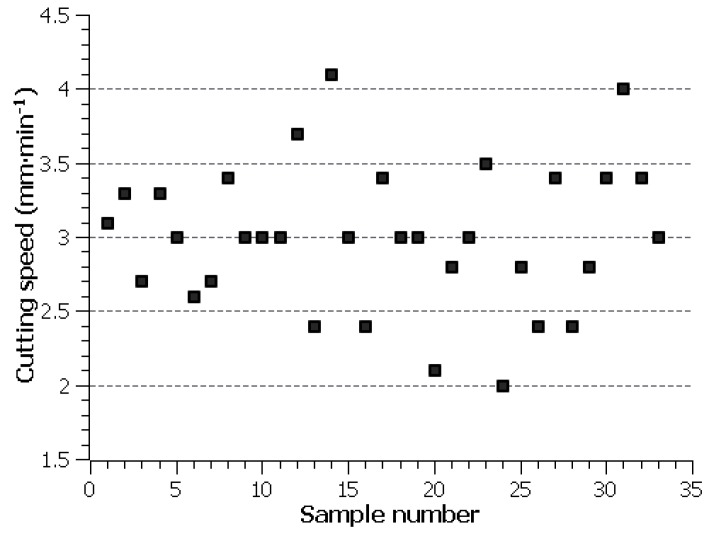
Cutting speed value of individual samples.

**Figure 3 materials-12-03758-f003:**
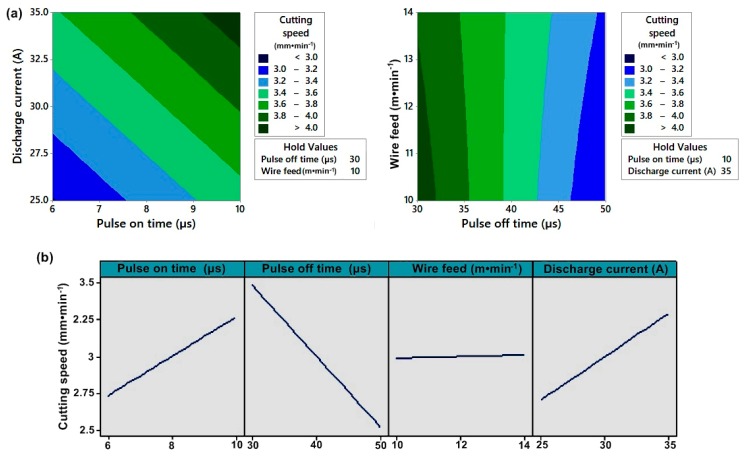
(**a**) Contour plot of the cutting speed; (**b**) Main effect plot for the cutting speed.

**Figure 4 materials-12-03758-f004:**
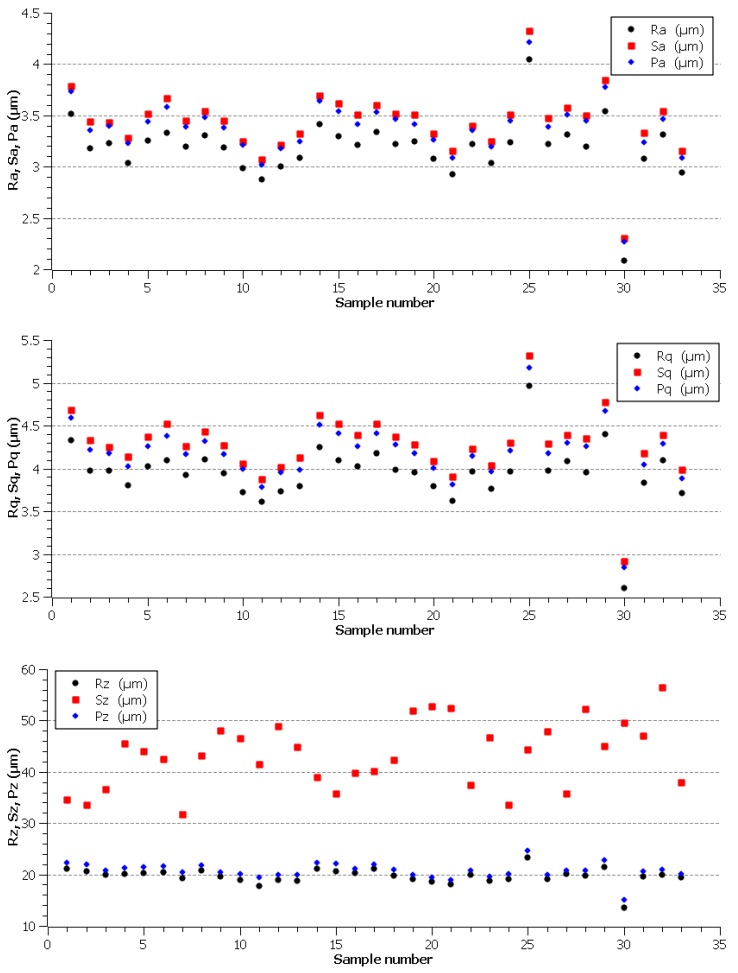
The evaluated parameters of the basic profile, profile and planar parameters of individual experimental samples.

**Figure 5 materials-12-03758-f005:**
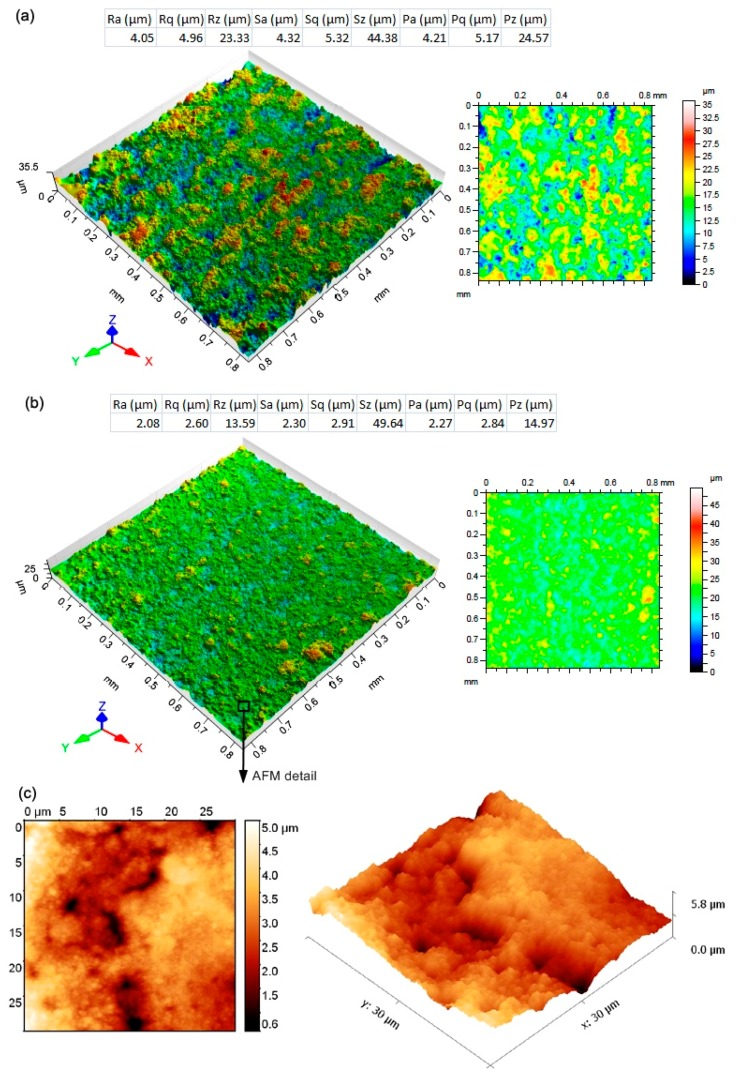
(**a**) the evaluated surface topography parameters of Sample 25, including color−filtered relief of its surface; (**b**) evaluated surface topography parameters of Sample 30, including a color−filtered relief of its surface; (**c**) 3D relief of Sample 30 from a given area obtained by AFM.

**Figure 6 materials-12-03758-f006:**
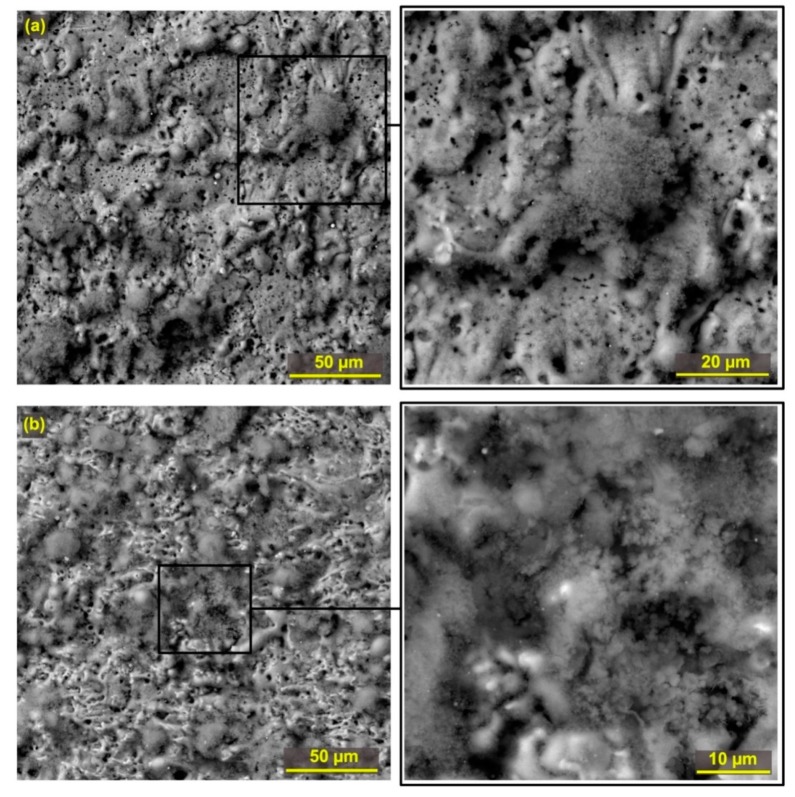
The sample surface morphology including magnified SEM (BSE) detail (**a**) Sample 30 machined with parameters: T_off_ = 30 µs, U = 70 V, T_on_ = 6 µs, I = 35 A, v = 14 m·min^−1^; (**b**) Sample 25 machined with parameters: T_off_ = 40 µs, U = 60 V, T_on_ = 8 µs, I = 30 A, v = 12 m·min^−1^.

**Figure 7 materials-12-03758-f007:**
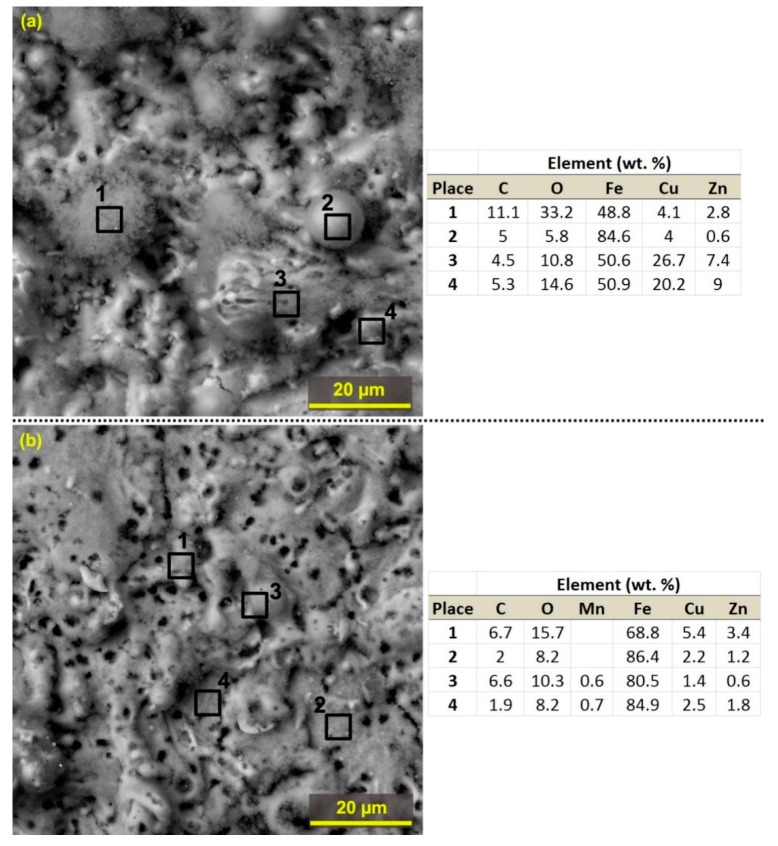
The sample surface morphology including magnified SEM (BSE) detail (**a**) Sample 25 machined with parameters: T_off_ = 40 µs, U = 60 V, I = 30 A, T_on_ = 8 µs, v = 12 m·min^−1^; (**b**) Sample 30 machined with parameters: T_off_ = 30 µs, U = 70 V, I = 35 A, T_on_ = 6 µs, v = 14 m·min^−1^.

**Figure 8 materials-12-03758-f008:**
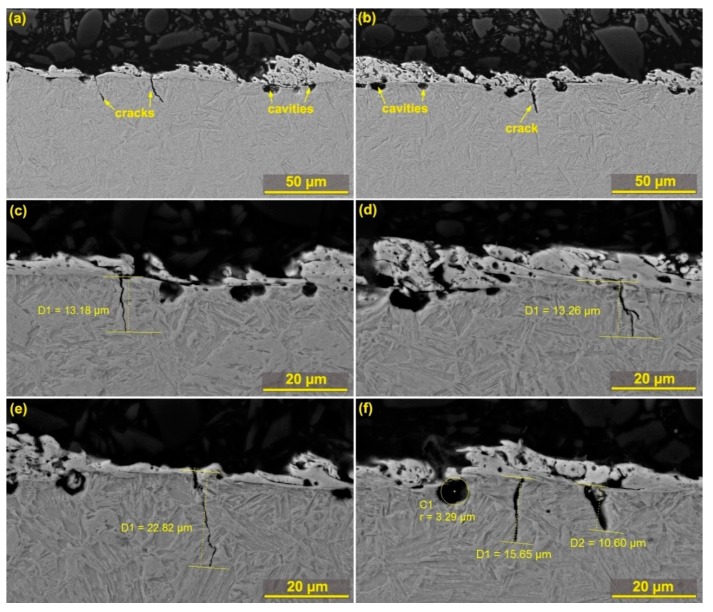
The cross sectional view showing subsurface SEM (BSE) defects (**a**) Sample 25 machined with parameters: T_off_ = 40 µs, U = 60 V, T_on_ = 8 µs, I = 30 A, v = 12 m·min^−1^; (**b**) Sample 1 machined with parameters: T_off_ = 40 µs, U = 70 V, T_on_ = 8 µs, I = 30 A, v = 12 m·min^−1^; (**c, d**) Sample 20 machined with parameters: T_off_ = 50 µs, U = 70 V, T_on_ = 6 µs, I = 25 A, v = 14 m·min^−1^; (**e, f**) Sample 13 machined with parameters: T_off_ = 50 µs, U = 70 V, T_on_ = 10 µs, I = 25 A, v = 10 m·min^−1^.

**Figure 9 materials-12-03758-f009:**
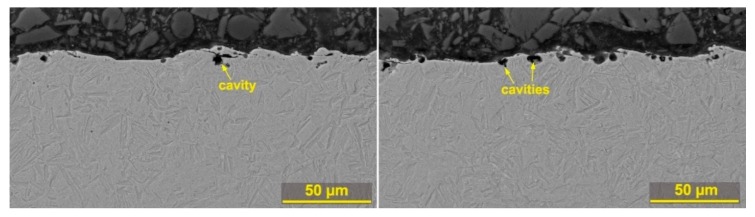
The cross section of Sample 30 machined with parameters: U = 70 V, T_on_ = 6 µs, T_off_ = 30 µs, v = 14 m·min^−1^ and I = 35 A, SEM (BSE), magnified at 1000×.

**Figure 10 materials-12-03758-f010:**
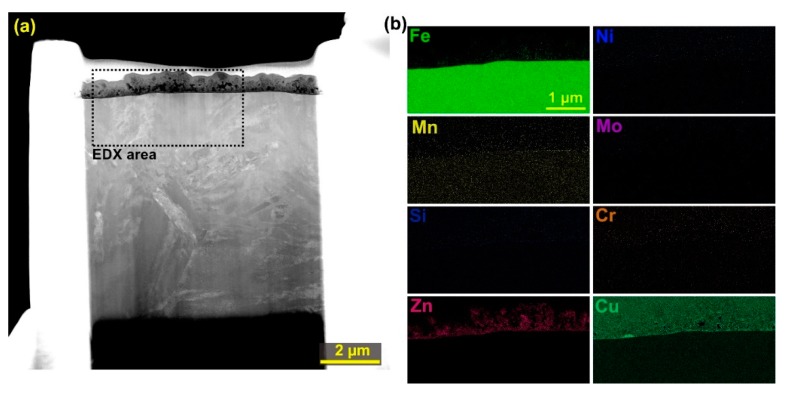
(**a**) TEM lamella; (**b**) maps of the distribution of individual elements in the examined area of the lamella.

**Table 1 materials-12-03758-t001:** The limit values of the machine setting parameters.

Parameter	Discharge Current (A)	Pulse off Time (µs)	Pulse on Time (µs)	Gap Voltage (V)	Wire Feed (m·min^−1^)
Level 1	25	50	6	50	10
Level 2	30	40	8	60	12
Level 3	35	30	10	70	14

**Table 2 materials-12-03758-t002:** Machining parameters used in the experiment.

Number of Sample	Discharge Current (A)	Gap voltage (V)	Pulse off Time (µs)	Pulse on Time (µs)	Wire Feed (m·min^−1^)	Number of Sample	Discharge Current (A)	Gap Voltage (V)	Pulse off Time (µs)	Pulse on Time (µs)	Wire Feed (m·min^−1^)
**1**	30	70	40	8	12	**18**	30	60	40	8	12
**2**	30	60	30	8	12	**19**	30	60	40	8	12
**3**	25	60	40	8	12	**20**	25	70	50	6	14
**4**	30	60	40	10	12	**21**	25	50	30	6	14
**5**	30	50	40	8	12	**22**	30	60	40	8	12
**6**	30	60	50	8	12	**23**	25	70	30	10	14
**7**	30	60	40	6	12	**24**	25	50	50	6	10
**8**	35	60	40	8	12	**25**	30	60	40	8	12
**9**	30	60	40	8	10	**26**	25	50	50	10	14
**10**	30	60	40	8	14	**27**	25	50	30	10	10
**11**	30	60	40	8	12	**28**	35	50	50	6	14
**12**	35	50	30	6	10	**29**	35	50	50	10	10
**13**	25	70	50	10	10	**30**	35	70	30	6	14
**14**	35	70	30	10	10	**31**	35	50	30	10	14
**15**	30	60	40	8	12	**32**	30	60	40	8	12
**16**	35	70	50	6	10	**33**	25	70	30	6	10
**17**	35	70	50	10	14	–	–	–	–	–	–

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
