# Peer review of "The Influence of WEDM Parameters Setup on the Occurrence of Defects When Machining Hardox 400 Steel"

_materials, 2019, doi:10.3390/ma12223758_

Round 1
Reviewer 1 Report
The paper (i.e. the study itself) is clear and well focused, the conclusion effectively wraps up and goes beyond restating the thesis. Writing is smooth, skillful and coherent. The sentences are strong and expressive with varied structure. The diction is consistent and words, figures ad tables are well chosen.
Depending on the parameters of materials treatment and the type and size of the introduced stresses from the technological procedures there can affect negatively or positively properties of steel. On basis of literature research and their own research the authors are presenting the main and interesting influence of the selected technological factors on the surface properties of hardox stell.
The article is written at a good level, I make no comments and recommend the article for publication without any changes.
Author Response
Dear reviewer,
I would like to thank you for your time and your positive comments.
Best regard
Mouralova
Reviewer 2 Report
This paper mainly analyzes the surface morphology of redox 400 after WEDM, analyzes the specific situation of the surface and sub surface in detail, and analyzes the impact of pulse cutting time on the cutting speed through the statistical evaluation of the cutting speed. The main opinions are as follows:
1. The title of the paper is to discuss the defects in the processing of 400 steel. The significance of the section of statistical evaluation of cutting speed is not explained in the article, which is a bit abrupt.
2. The concept of "Introduction" needs to be improved. Can we increase the properties of some 400 steel and its influence degree.
3. The details of the experiment are insufficient. There is no explanation for the choice of "half response surface design" in the experiment.
4. There is a lack of a detailed discussion in the analysis and discussion part of the paper to explain the exact significance of your findings.
5. In Figure 10 (b) of this paper, I don't know how to see the changes of matrix elements.
6. The setting parameters the arc on time is grammatically incorrect. "within micrometres" should be "within microns".
Author Response
This paper mainly analyzes the surface morphology of redox 400 after WEDM, analyzes the specific situation of the surface and sub surface in detail, and analyzes the impact of pulse cutting time on the cutting speed through the statistical evaluation of the cutting speed. The main opinions are as follows:
1. The title of the paper is to discuss the defects in the processing of 400 steel. The significance of the section of statistical evaluation of cutting speed is not explained in the article, which is a bit abrupt.
I added to the previous chapter: In the following chapter, the optimization of machine setting parameters was made in order to maximize cutting speed in order to increase machining efficiency in the form of reduced machining time.
The concept of "Introduction" needs to be improved. Can we increase the properties of some 400 steel and its influence degree.
I added properties of Hardox to the Introduction chapter.
The details of the experiment are insufficient. There is no explanation for the choice of "half response surface design" in the experiment.
This type of experiment was used for its adequacy with respect to the number of actually produced and evaluable samples, all due to the financial demands of the whole experiment and all subsequent extensive analyzes.
There is a lack of a detailed discussion in the analysis and discussion part of the paper to explain the exact significance of your findings.
Unfortunately, the discussion was limited. The machining of this abrasion-resistant steel with WEDM has not been studied in any study yet.
In Figure 10 (b) of this paper, I don't know how to see the changes of matrix elements.
It is based on the amount of colored points of each element. Maybe it is not good visible in resolution of this picture, because of limited size of picture on paper page A4. Im sorry for that.
The setting parameters the arc on time is grammatically incorrect. "within micrometres" should be "within microns".
I fixed it. Thank you
Thank you very much for your time and suggestive comments, which I highly appreciate.
With regards
Katerina Mouralova